# Bioactive Compounds from BRS Violet Grape Pomace: An Approach of Extraction and Microencapsulation, Stability Protection and Food Application

**DOI:** 10.3390/plants12183177

**Published:** 2023-09-05

**Authors:** Edilson Bruno Romanini, Leticia Misturini Rodrigues, Ana Paula Stafussa, Talita Perez Cantuaria Chierrito, Aline Finger Teixeira, Rúbia Carvalho Gomes Corrêa, Grasiele Scaramal Madrona

**Affiliations:** 1Postgraduate Program in Food Science, Universidade Estadual de Maringá, Avenida Colombo 5790-Zona 7, Maringá 87020-900, PR, Brazil; brunoromanini84@gmail.com (E.B.R.); leticia_misturini@hotmail.com (L.M.R.); anastafussa@gmail.com (A.P.S.); 2Instituto Federal do Paraná, Campus Paranavaí, Avenida José Felipe Tequinha, 1400-Jardim das Nacoes, Paranavaí 87703-536, PR, Brazil; aline.finger@ifpr.edu.br; 3School of Pharmaceutical Sciences of Ribeirão Preto, University de São Paulo, Avenida do Café, Ribeirão Preto 14040-903, SP, Brazil; talitacantuaria@yahoo.com.br; 4Postgraduate Program in Clean Technologies, Cesumar University-UNICESUMAR, Maringá 87050-390, PR, Brazil; rubia.correa@unicesumar.edu.br; 5Cesumar Institute of Science, Technology and Innovation-ICETI, Maringá 87050-390, PR, Brazil; 6Department of Food Engineering, State Universidade Estadual de Maringá, Avenida Colombo 5790-Zona 7, Maringá 87020-900, PR, Brazil

**Keywords:** freeze-drying, gelatin, maltodextrin, xanthan gum, anthocyanins

## Abstract

Microencapsulating phenolic compounds and anthocyanins from grape pomace, a by-product of the food industry, is attractive because of the many beneficial health effects associated with these compounds. At first, we evaluated the cultivar BRS Violeta using microencapsulation, indicating the degree of innovation in the present research. This study aims to microencapsulate grape pomace extract in a combination of maltodextrin and xanthan gum via lyophilization, and determine the protective effect of this microcapsule on the phenolic compounds and anthocyanins. Thus, the microcapsule stability was determined over 120 days, under different temperature conditions (4 and 25 °C) and in the presence or absence of light. Additionally, a gelatin application test was performed to investigate the effect of the microcapsule on color stability. When comparing the extract versus microcapsules, the microcapsule results were better both for total anthocyanins (1.69 to 1.54-fold) and total phenolic compounds (3.06 to 1.74-fold), indicating a longer half-life after encapsulation. The microcapsule application in gelatin demonstrated that the encapsulating matrix retained the color for 30 days. Thus, the encapsulation method can be recommended to preserve the bioactive compounds and the coloration in food products such as gelatin.

## 1. Introduction

Since studies have shown a relationship between free radicals and the development of some diseases linked to oxidative stress, compounds that can neutralize these particles have gained importance and have been more extensively studied for their application in human health [1]. In nature, some of these substances are found in vegetables, in which they are responsible for adaptation to the environment. Among these phytochemicals, phenolic compounds are very important because of their antioxidant activity. These compounds can be divided into two groups: flavonoids (anthocyanins, flavonols, flavano-3-ols, and condensed tannins) and non-flavonoid compounds (stilbenes and phenolic acids) [2].

A large amount and variety of phenolic compounds are found in grapes, especially the anthocyanins responsible for their color. As stated earlier, these substances have antioxidant properties related to the prevention of inflammatory, allergic, cancer, neurodegenerative, and cardiovascular diseases [1,3].

Given the concern regarding health and food safety, to encourage the use of products of a natural origin instead of antioxidants of a synthetic origin, the extraction of these substances from natural substrates is increasing. In the case of grapes, it is perfectly possible to extract the phenolic compounds from the pomace generated by the wine industry after pressing [4]. After extraction, these substances can be applied in the food, pharmaceutical, and cosmetic industries [5,6].

Nowadays, the greatest difficulty is to use these phenolic compounds in processes that maintain their properties for a reasonable time after grape extraction, for example, since these molecules, when exposed for a long time, are unstable and can easily degrade, oxidize, epimerize and polymerize. The chemical structure of the bioactive compounds that cause the antioxidant action is the result of enzymes and radiation, as well as variations in temperature, oxygen, and pH [7,8].

In this circumstance, the application of substances capable of transporting these polyphenols and providing protection until their final destination has been mentioned in the literature, such as encapsulation technologies that use lyophilization and spray drying [9,10]. Microencapsulation is a technique that can be applied to incorporate and immobilize a biologically active compound (for example, phenolic compounds) in or on solid particles or liquid vesicles to stabilize and protect the biologically active compound. Microencapsulation is widely used when it is desirable to maintain the stability and improve the shelf life of molecules sensitive to light and heat, such as anthocyanins [11,12].

Among the various microencapsulation agents, maltodextrins (MD) are widely used as a drying agent and are proven to be non-toxic sugar polymers, as they have the solubility and produce the viscosities expected for a good encapsulating agent, in addition to their low cost and the absence of any influence on the taste and odor of the final product. They also have advantages such as delayed crystallization, reduced total dissolved solids, reduced acidity loss, reduced pH loss, and reduced hygroscopicity of the mixture [13,14].

Another natural-origin encapsulating agent that is studied is the high-molecular-weight anionic biopolymer of xanthan gum, which is colorless, odorless, tasteless, and stable at higher temperatures and at pH variations [15]. 

This study aimed to report the extraction and microencapsulation, stability protection, and food application of grape pomace. For this, the aqueous extract rich in polyphenols from the grape (cultivar BRS Violeta) pomace of the juice industry was encapsulated in a mixture of maltodextrin (MD) and xanthan gum (GX) (9.5:0.5) and the stability of the encapsulated content was observed for 120 days. Additionally, an application test in gelatin was carried out to monitor the microcapsule’s effect on maintaining the extract’s color. As far as we know, there are no records in the literature of similar studies with the cultivar BRS Violeta, thus indicating the degree of innovation of the present research.

## 2. Results and Discussion

### 2.1. Microcapsule Stability and Encapsulation Efficiency (EE) 

The EE for the MD/GX was 68.10%. These results were better than those observed by Antigo et al. [16], who obtained an EE of 43.45% using beet extract and encapsulation via lyophilization with MD/GX. The study by Mahdavi et al. [17], using barberry extract and encapsulation via spray drying with three different wall materials—that is, combining maltodextrin, gum arabic, maltodextrin, and gelatin—obtained higher values (ranging from 89.06 to 96.21%) when compared with the ones observed in the present study. In the study conducted by Idaham et al. [18], the researchers achieved a high encapsulation efficiency of 99.69% when microencapsulating an anthocyanin extract from hibiscus using a combination of maltodextrin and gum arabic as the encapsulating materials. Similarly, Silva et al. [19] obtained encapsulation efficiencies of 83.21% and 99.02% when microencapsulating jabuticaba extracts. Maltodextrin and gum arabic are commonly used as coating materials for microencapsulating anthocyanins and other bioactive compounds due to their effectiveness in protecting the compounds from degradation, and improving their solubility and stability in food and beverage applications. These data demonstrate that the EE may be related to the composition of the encapsulating agent, which confers different physicochemical properties for film formation, as well as the drying method (lyophilization or spray drying, for example) [20].

The stability of the total anthocyanins (TA) and total phenolics (TP) in the powdered freeze-dried microcapsules and control samples (extract) were evaluated under different storage conditions (Table 1).

Overall, the encapsulated content stability assay showed a higher degradation with the extract (control) and when exposed to light and temperature, both for TP and TA, as expected, and as demonstrated by Rodrigues et al. [21] (Table 1). Some authors also mention that the use of encapsulating agents in adequate proportions can guarantee the protection of the active substance and the preservation of its functionality [22].

The data indicate that there is a protective effect of the microcapsule on TA and TP compared to the non-encapsulated extract (control), which demonstrated a considerable loss of TA and TP over 120 days, demonstrating an even higher degradation in the presence of light (20.61% and 24.49% for TA and TP, respectively) while the loss when encapsulation was performed was 14.17% and 17.92% for TA and TP. In addition, the effect of time was essential for the degradation of both TA and TP. Table 1 shows a significant difference between all samples when comparing the beginning (0 days) and the end of the analysis (120 days).

When verifying the effect of temperature on TA at 4 °C and 25 °C, the encapsulated content was not altered over the 120 days. For TP, there was a loss of 9.69% at 4 °C and 12.06 at 25 °C. Idham et al. [18], using spray drying and different encapsulating matrices for anthocyanins, visualized that the storage time and the encapsulating agent considerably altered the color, while the storage temperature did not prove to be an influencing factor. Differing from this report, in another study using maltodextrin and dextrose as encapsulating agents, the anthocyanin content of encapsulated black carrot extract decreased by 11% and 33% over 64 days of storage at 4 and 25 °C, respectively [23]. 

In this context, the present study found that the samples stored at temperatures of 25 °C lost more phenolic compounds and anthocyanins compared to those stored at 4 °C. However, the presence of the encapsulation system protected against temperature-induced degradation. Without encapsulation, the degradation of these compounds could have been even more pronounced, highlighting the crucial role of encapsulation in reducing the adverse effects of temperature on phenolic compounds.

Ravichandran et al. [24] extracted betalains from beetroot and encapsulated them in different matrices (maltodextrin, guar gum, gum arabic, pectin, and xanthan gum) and at different concentrations, and the samples were dried via lyophilization or spray drying. The lyophilized MD/GX sample showed better stability between the conditions analyzed, corroborating the results obtained in the present study.

When comparing the presence and absence of light during storage at 25 °C, concentrations decreased in both cases for TA (14.17% with light and 12.78% without light) and TP (17.92% with light and 12.06% without light). The TP was more degraded than the TA, indicating that the presence of light influenced the degradation of this group of compounds, as already reported elsewhere [4,16,23]. Burin et al. [25] studied the effects of temperature and exposure to light on the stability of anthocyanins from Cabernet Sauvignon, and their results also revealed that light is a significant factor in accelerating the degradation of anthocyanins.

At 120 days of storage, the smallest color change observed was for Cap4 (ΔE < 5). The extracts, regardless of their storage conditions (temperature, with and without light) showed the greatest color variations (ΔE > 5). The protective effect of MD/GX was also compared to the extract by Rodrigues et al. [21], whose jabuticaba extract, stored for 36 days under the same conditions evaluated in the present study, was more degraded than the encapsulated samples. According to Romero-González et al. [26], using three polysaccharides (maltodextrin, gum arabic, and inulin) as encapsulating agents and lyophilization for drying, a value of ΔE < 27 was obtained, even after 60 days of storage at 25 °C, attributing this result of color stability to low water activity.

Due to the best results presented by Cap4 for the color and the TA and TP contents, this sample was applied in a food matrix (gelatin) to evaluate the color stability during storage for 30 days. Gelatin was chosen because it can be stored in the dark at 4 °C.

Considering that the present research focuses on evaluating the encapsulated content, Figure 1 presents the specific data during storage. When verifying the behavior of the three parameters (an absence of light at 4 °C, and an absence and presence of light at 25 °C) on the stability of the encapsulated content after 120 days, a significant difference was observed in the TP content, as there was an increased degradation in the presence of light. The evaluation of the stability of anthocyanins subjected to a temperature of 25 °C in the absence and presence of light, and to a temperature of 4 °C, showed that the samples were not significantly different. However, after 120 days, the TA contents differed significantly, resulting in a loss of approximately 13–14%. The encapsulation process can improve the stability of anthocyanins against light and temperature (Figure 1).

Table 2 demonstrates the degradation kinetics and half-life of the encapsulated contents evaluated during the storage period (120 days).

The encapsulated contents showed a first-order degradation curve for all samples (Table 2), corroborating previous reports. Temperature is a critical factor in the stability of encapsulated substances. The results demonstrated that the degradation rate constant (K) was higher for samples kept at 25 °C compared to those stored at 4 °C. This indicates that the rate of degradation increases at higher temperatures, consistent with the kinetic theory of chemical reactions. The samples’ half-lives (t_(1/2)_) decreased with increasing temperature, indicating a reduction in the shelf life at higher temperatures. First-order kinetics was used for the degradation of anthocyanins extracted from blueberries [27], from hibiscus stored at temperatures from 5 °C to 30 °C for 50 days [28], and from wine residues with a long-term stability of 6 months in the absence and presence of light [11]. 

When analyzing the encapsulated sample versus the extract (control) at both temperatures, a decrease in the TP content present in the extract was observed. This is evidenced by the t_(1/2)_ of 984 and 221 days at 4 °C for the microcapsules and the extract, respectively. When analyzing the effect of light on the microcapsules and the extract, there were similar losses for TA (t_(1/2)_ of 621 and 367 days) and for TP (495 and 283 days, respectively, for the microcapsules and the extract). With the results presented, it can be seen that, under the conditions studied, anthocyanins and phenolic compounds showed an increase in degradation in the extract, demonstrating that the microcapsule promotes the protection of the bioactive compounds. The stability of encapsulated substances can be considerably influenced by extrinsic elements. Therefore, when developing encapsulated products for practical applications, it is crucial to carefully optimize storage conditions to ensure the maintenance of product quality and effectiveness over time.

### 2.2. Microcapsule Application in Gelatin

Analyzing the changes in gelatin color (ΔE) during storage is important for understanding the stability of this product under different conditions. Measuring the ΔE provides an objective assessment of the noticeable changes in color over time. 

Colorless gelatin was used as the base in which the dye, the extract, and the microcapsules were applied (Figure 2) to study the variation in the color of the sample during storage at 4 °C for 30 days.

The use of MD/GX provided a significant increase in luminosity (L*), from 37.65 (raw extract) to 44.00 (encapsulated extract). The dry microcapsules were brighter than the dry extract, probably due to the white coloration of maltodextrin, which has already been mentioned in other studies [29].

Color differences from 0 to 1.5 could be judged as small and nearly equal for visual analysis. They could be distinguished in the range of 1.5 to 5, while color differences were only evident for a ΔE higher than five [30].

The total color of the microcapsules and the dye were similar after 30 days (ΔE < 5). The extract showed a color change (ΔE > 5) during storage (days 15 and 30) that was 2.47 times higher than the dye and 2.65 times higher than the microcapsules after 30 days, indicating the fragility of this compound according to the previous analyses (Table 1 and Table 2), even when added to a complex food matrix (such as gelatin). Rodrigues et al. [21] encapsulated jabuticaba bark extract in maltodextrin and xanthan gum using lyophilization and applied it in a gelatin matrix to analyze the color stability for 60 days at 25 °C. The extract alone presented the highest color variation (ΔE > 5), while the incorporation of microcapsules into the gelatin had the smallest change (ΔE < 5). Moser et al. [31] evaluated the color stability of BRS Violeta red grape juice powder that was encapsulated (maltodextrin and proteins) via spray drying during storage for 150 days at 35 °C and, regardless of the type of carrier applied, the temperature and storage time did not change the color of the reconstituted grape juice. Differences (ΔE) were lower than 1.8 for any time/temperature combination. In another study, a powdered food dye was obtained via spray drying *Opuntia stricta* fruit juices, with the application of glucose syrup to contribute to the drying process. The dye was tested in two food models, yogurt and soft drinks, and stored under refrigeration (4 °C) for one month, and there was no visible difference in the color (ΔE < 5) [30]. 

These results strongly suggest that the combination of MD/GX as an encapsulating agent protects colored substances from natural extracts, and that it can be applied as a substitute for artificial dyes in food products such as gelatin when stored for up to 30 days under refrigeration.

## 3. Materials and Methods

### 3.1. Raw Material and Reagents

The residue used from the grape variety (BRS Violeta) came from the city of Marialva (PR, Brazil). The extract (E) was obtained using the ultrasound-assisted method, and water was used as a solvent in the ratio of 1:200 (*w*/*v*, g/mL) with a total extraction volume of 200 mL, for 6 min at 55 °C with an amplitude of 40%, performed as described in the study previously published by Romanini et al. [4]. The maltodextrin (DE 10) was provided by Cargill (SP, Brazil) and the xanthan gum was purchased from a local store in Maringá (PR, Brazil). The reagents used in the work were of analytical grade.

### 3.2. Extract Encapsulation and Encapsulation Efficiency (EE)

The encapsulating agents were added at an agent–extract ratio of 1:1 (*w*/*w*), according to Ferrari et al. [32]. Maltodextrin and xanthan gum were added at concentrations of 99.5% and 0.5%, respectively, as described by Antigo et al. [16] and Cruz et al. [22]. The microcapsules containing the extract and the raw extract were frozen for 48 h at −18 °C and then lyophilized (freeze L108, Liobras, São Carlos, São Paulo, Brazil) for 2 days for maximum drying. 

The encapsulation efficiency was calculated according to Cabral et al. [33], using water as a solvent, according to Equation (1)
EE (%) = ((FT − FS) × 100)/FT (1)
where FT and FS refer to the total phenolic compounds inside the microcapsules (initial) and on the surface (extracted).

### 3.3. Analysis of the Bioactive Compounds of the Extracts 

The total anthocyanin content (TA) was determined using the pH differential absorbance method at 520 and 700 nm and pH 1.0 and pH 4.5 [34]. The total anthocyanin content was expressed in mg of cyanidin-3-glucoside/g of grape pomace. The Folin–Ciocalteu method was used for the determination of the total phenolic compounds (TP). The total phenolic compounds were measured with the calibration standard curve using gallic acid, and the values were expressed in mg of gallic acid equivalent (GAE)/g of grape pomace [35,36].

### 3.4. Encapsulated Content Stability during Storage

After lyophilization, the microcapsules containing the extract were weighed and then stored in transparent plastic packages in a chamber (BOD Incubator) at 25 °C in the presence and absence of light, using two 20 W fluorescent lamps (presence of light) and a dark chamber (absence of light), and at 4 °C, for 120 days [37]. The monomeric total anthocyanins, total phenolic compounds, and color parameters were monitored. As a control, the lyophilized extract was used. 

The following samples and storage conditions were obtained and evaluated: Cap4 = encapsulation at 4 °C without light; Ext4 = extract at 4 °C without light; Cap25 = encapsulation at 25 °C without light; Ext25 = extract at 25 °C without light; CapLight = encapsulation in the presence of light at 25 °C; and ExtLight = extract in the presence of light at 25 °C.

The anthocyanin and total phenolic compounds experimental data values were analyzed using the first-order reaction degradation kinetics and half-life (t_1/2_) methods [38], according to Equations (2) and (3)
ln (A_t_/A_0_) = −k × t(2)
t_1/2_ = ln 2/k(3)
where t is time (day), A_t_ is the level of TA or TP, A_0_ is the initial TA or TP level at time zero, and k is the rate constant (day^−1^).

The percentage of the total loss of anthocyanins and the total phenolic compounds over the storage period was determined by the ratio between the concentration on the last day of storage (d120) and the initial concentration (d0), as mentioned in Souza et al. [39].

### 3.5. Gelatin Application

The microcapsules, the extract, and an artificial dye (Gran chef, easy gel—burgundy color, Brazil) were used with a colorless gelatin powder (Dr. Oetker^®^, Bielefeld, Germany). The preparation of pure gelatin followed the manufacturer’s instructions. Then, the powdered samples were placed together with the gelatin solutions and manually stirred until homogenization occurred. The proportions of microcapsule, extract, and dye for dissolved gelatin were from 1.8% to 98.2%, as described by Rodrigues et al. [21]. Subsequently, the samples were placed in Petri dishes and randomly stored at 4 °C until analysis, simulating consumption conditions. The gelatin samples were refrigerated (4 °C) for 30 days, and the color parameters were analyzed at 0, 15, and 30 days.

### 3.6. Color Parameters

The color of the samples was evaluated with a portable colorimeter CR-400 (Minolta Ltd., Osaka, Japan), with an illuminant D65-angle of 10°. The luminosity parameters of L* black to white, a* green to red, and b* blue to yellow were verified. The color difference (ΔE) was determined using Equation (4), according to the method suggested by Silva et al. [40]
ΔE = √ (L*dayX − L*day0)^2^ + (a*dayX − a*day0)^2^ + (b*dayX − b*day0)^2^(4)
where L*, a* and b* = color parameters; day0 = initial day; and dayX = 15 and 30 days of storage, respectively.

### 3.7. Data Analysis

All readings were carried out in triplicate and performed to analyze the variance, and Tukey’s test was used to calculate the minimum significant difference (*p* < 0.05) between the mean values. Excel and Statistic 7.0 software were used for this analysis.

## 4. Conclusions

The use of maltodextrin and xanthan gum as encapsulating materials effectively preserved the stability of phenolic compounds and anthocyanins in grape pomace extract during storage for 120 days and under different conditions (temperature, and absence and presence of light), resulting in an increased half-life compared to the isolated extract.

MD/GX encapsulation was used to evaluate color changes in gelatin. The encapsulated content maintained its initial color after 30 days of refrigerated storage with a ΔE measurement of less than five.

These results are important to expand the use of encapsulated natural dyes, enabling the application of these compounds in the food, pharmaceutical, and cosmetic industries.

## Figures and Tables

**Figure 1 plants-12-03177-f001:**
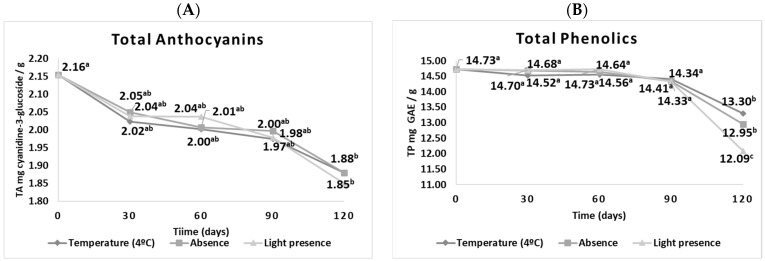
Degradation of the total anthocyanin (**A**) and phenolics (**B**) of the microcapsules over 120 days of storage at low temperature (4 °C) and at room temperature (25 °C) in the absence or presence of light. Different letters indicate a significant difference when different times are compared (*p* ≤ 0.05).

**Figure 2 plants-12-03177-f002:**
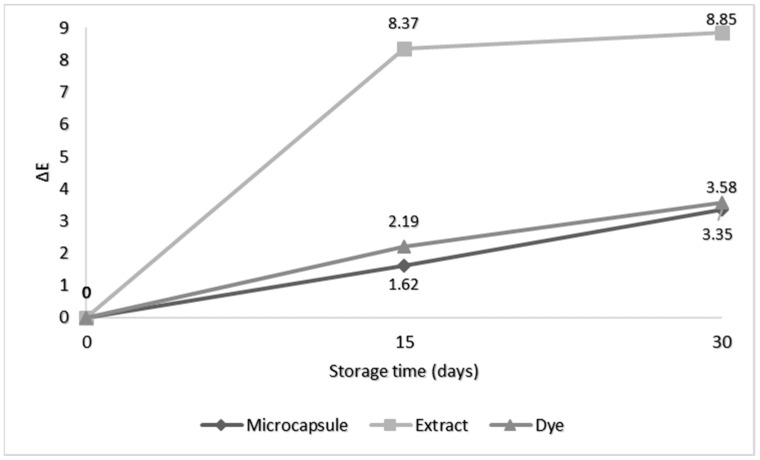
Variation in the color of the gelatin (ΔE) during the 30 days of storage at 4 °C.

**Table 1 plants-12-03177-t001:** Total phenolic compounds (TP), total anthocyanins (TA), and delta E during 120 days of storage in the presence/absence of light (at 25 °C), and at 4 and 25 °C (in the dark).

Stability (Days)	Cap4	Ext4	Cap25	Ext25	CapLight	ExtLight
	**TP (mg GAE/g)**					
0	14.73 ^a *^ ± 0.21	46.17 ^a^ ± 0.04	14.73 ^a^ ± 0.21	46.17 ^a^ ± 0.04	14.73 ^a^ ± 0.21	46.17 ^a^ ± 0.04
30	14.52 ^a^ ± 0.11	44.33 ^b^ ± 0.13	14.68 ^a^ ± 0.07	46.51 ^a^ ± 0.59	14.7 ^a^ ± 0.12	45.36 ^a^ ± 0.42
60	14.56 ^a^ ± 0.10	43.82 ^b^ ± 0.31	14.64 ^a^ ± 0.03	42.46 ^b^ ± 0.10	14.73 ^a^ ± 0.04	43.71 ^a^ ± 0.47
90	14.41 ^a^ ± 0.16	39.41 ^c^ ± 0.22	14.34 ^a^ ± 0.18	40.35 ^c^ ± 0.50	14.33 ^a^ ± 0.17	38.19 ^b^ ± 0.24
120	13.30 ^b^ ± 0.07	35.42 ^d^ ± 0.38	12.95 ^b^ ± 0.13	36.30 ^d^ ± 0.74	12.09 ^b^ ± 0.08	34.86 ^c^ ± 0.46
Loss (%)	9.69	23.27	12.06	21.38	17.92	24.49
	**TA (mg cyanidine-3-glucoside/g)**					
0	2.16 ^a^ ± 0.02	5.30 ^a^ ± 0.04	2.16 ^a^ ± 0.02	5.30 ^a^ ± 0.04	2.16 ^a^ ± 0.02	5.30 ^a^ ± 0.04
30	2.02 ^b^ ± 0.04	5.21 ^a^ ± 0.11	2.05 ^ab^ ± 0.15	5.08 ^ab^ ± 0.11	2.04 ^ab^ ± 0.04	5.19 ^a^ ± 0.11
60	2.00 ^b^ ± 0.03	5.15 ^a^ ± 0.03	2.01 ^ab^ ± 0.09	5.03 ^ab^ ± 0.02	2.04 ^ab^ ± 0.03	4.91 ^b^ ± 0.03
90	1.97 ^b^ ± 0.01	4.88 ^b^ ± 0.11	2.00 ^ab^ ± 0.03	4.82 ^c^ ± 0.06	1.98 ^ab^ ± 0.12	4.67 ^c^ ± 0.10
120	1.88 ^c^ ± 0.08	4.27 ^c^ ± 0.03	1.88 ^b^ ± 0.02	4.32 ^d^ ± 0.10	1.85 ^b^ ± 0.18	4.21 ^d^ ± 0.03
Loss (%)	12.78	19.47	12.78	18.53	14.17	20.61
	**Delta E**					
120	4	11	6	10	8	12

Mean values ± standard deviation (n = 3 repetitions). The time difference was evaluated by % of loss. * Different letters in the same column indicate a significant difference in terms of days (*p* ≤ 0.05). GAE = gallic acid equivalent; Cap4 = encapsulation at 4 °C; Ext4 = extract at 4 °C; Cap25 = encapsulation at 25 °C in the dark; Ext25 = extract at 25 °C in the dark; CapLight = encapsulation in the presence of light; and ExtLight = extract in the presence of light.

**Table 2 plants-12-03177-t002:** Kinetics of the degradation rate constant (K), and half-life of the encapsulated contents and the extract during storage in the presence/absence of light, and at 4 and 25 °C (in the dark).

	Total Anthocyanin		Total Phenolics	
	K	t_(1/2)_ (Days)	K	t_(1/2)_ (Days)
**Cap4**	0.0010	698	0.0007	984
**Ext4**	0.0017	417	0.0022	321
**Cap25**	0.0010	694	0.0009	741
**Ext25**	0.0015	450	0.0021	334
**CapLight**	0.0011	621	0.0014	495
**ExtLight**	0.0019	367	0.0024	283

K = degradation rate constant; t_(1/2)_= half-life Cap4 = microcapsules at 4 °C in the dark; Ext4 = extract at 4 °C, in the dark; Cap25 = microcapsules at 25 °C, in the dark; Ext25 = extract at 25 °C, in the dark; CapLight = microcapsules at 25 °C, in the presence of light; and ExtLight = extract at 25 °C, in the presence of light.

## Data Availability

The datasets supporting the conclusions of this article are included within the article.

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
