# Peer review of "Bioactive Compounds from BRS Violet Grape Pomace: An Approach of Extraction and Microencapsulation, Stability Protection and Food Application"

_plants, 2023, doi:10.3390/plants12183177_

Round 1

Reviewer 1 Report

Manuscript entitled "Bioactive compounds from BRS Violet grape pomace: New insights into extraction and microencapsulation, stability protection and food application" presents the method of encapsulating grape extract. The work is written in a logical way. I have a few comments:

1. The bioactivity of extracts was analyzed in the work, why were only the methods of determination of total phenolic compounds and total anthocyanins content selected? It is also worth doing the flavonoid content and antioxidant properties.

2. Section 2.2. Microcapsules application in gelatin - The color should be better described. Mainly ΔE is evaluated, what about other parameters, why are they not described?

Author Response

  1. The bioactivity of extracts was analyzed in the work, why were only the methods of determination of total phenolic compounds and total anthocyanins content selected? It is also worth doing the flavonoid content and antioxidant properties.

It was decided to work with these parameters at first in order to have an overall value, since the analysis of phenolic compounds indicates the value of all the compounds present (including flavonoids and phenolic acids), and within flavonoids, anthocyanins stand out in grapes, so it was analyzed. Results obtained in preliminary work, and data reported in the literature indicate that anthocyanins are not very stable making it important to evaluate them and that microencapsulation (which is the main objective of this work) is the process indicated in this case.

  1. Section 2.2. Microcapsules application in gelatin - The color should be better described. Mainly ΔE is evaluated, what about other parameters, why are they not described?

All the parameters were used to calculate delta E. However, considering that it indicates the total difference in color between the three coordinates (L, a* and b*) and that the objective was to evaluate stability in relation to the beginning and end of storage, it was decided to present it in this way.

Reviewer 2 Report

Thanks for the opportunity to review this research. The manuscript entitled Bioactive compounds from BRS Violet grape pomace: New in-sights into extraction and microencapsulation, stability protec-tion and food application” have described the extraction and microencapsulation, stability protection and food application using grape pomace. The subject of the manuscript is topical, but I recommend the publishing of the paper after the necessary corrections.

1. The abstract should be beginning with a sentence about the background of concept and the aims as well as novelty of study should be mentions. I recommend avoiding abbreviations in the abstract. Please improve.

3. Introduction: Check and format the citations in the whole manuscript. Also, Appropriate references must be provided to explained the background, what is already done and why this study carried out. Тhe phenolic composition of the grape pomace and the content of the other biologically active substances should be given more specifically. The scientific style should be used.

4. Material and methods: The used methods are accurate.

5. Results and discussion: The results are clearly presented. General remark to the discussion: In my opinion, the discussion provided by authors is difficult to follow and verify due missing critical details.

I have only one recommendation to authors: please improve the conclusion of the manuscript and check the text for technical errors.

Author Response

  1. The abstract should be beginning with a sentence about the background of concept and the aims as well as novelty of study should be mentions. I recommend avoiding abbreviations in the abstract. Please improve.

The abstract was improved.

  1. Introduction: Check and format the citations in the whole manuscript. Also, Appropriate references must be provided to explained the background, what is already done and why this study carried out. Тhe phenolic composition of the grape pomace and the content of the other biologically active substances should be given more specifically. The scientific style should be used.

The manuscript was checked.

  1. Material and methods: The used methods are accurate.

We really appreciate the positive feedback.

  1. Results and discussion: The results are clearly presented. General remark to the discussion: In my opinion, the discussion provided by authors is difficult to follow and verify due missing critical details.

I have only one recommendation to authors: please improve the conclusion of the manuscript and check the text for technical errors.

The conclusion was improved and was checked the text for technical errors.

Reviewer 3 Report

  The manuscript "Bioactive compounds from BRS Violet grape pomace: New insights into extraction and microencapsulation, stability protection and food application” by the authors Edilson Bruno Romanini, Leticia Misturini Rodrigues, Ana Paula Stafussa, Talita Perez Cantuaria Chierrito, Aline Finger Teixeira, Rúbia Carvalho Gomes Corrêa and Grasiele Scaramal Madrona, is dealing with the analysis of the possibility of applying microencapsulation of  grape pomace to obtain the bioactive product or colorant for food. 

It is a very important topic as stable preparation of bioactive compounds can be  a valuable  functional supplement for food.

 The topic is very interesting  but  presented manuscript data is well known and no new information is given to reader. The methods are very simple and traditional (Folin-Ciocalteu for TP and pH differential absorbance method), and some are weak described (EE).

As a simple report about influence of one of the mikroencapsulation method  (lyophilisation) on phenolics stability, manuscript can be considered for publication in Plants. I leave it to the editor's decision.

 In my opinion the work is suitable for publication after minor revision. 

My remarks are summarized as follows:

  • The title promises more than the manuscript actually contains. There is one extraction method and one microencapsulation method. The authors focused more on the stability of the extract during storage under different conditions. I suggest changing the title to a more appropriate one.

  • Microencapsulation consists of closing the biologically active compound in the biopolymer, so that it is protected from light, oxygen, water or other environmental factors. The advantage of microencapsulation lies in that the core material is completely coated and isolated from the external environment. Did the authors have the opportunity to verify that the coating material actually coated the particles of the extract? What is the opinion of the authors on the distribution of phenolic compounds in the microcapsule and how they determined the content of phenolic compounds inside and outside the capsule to determine EE?

Line 26 – TA and TP – explanation (total anthocyanins and total phenolics) should be given in first use 

Line 82 -GX -explanation (xanthan gum) should be given in first use 

Line 98 -“d=days” should be removed because in table  the word “days” is used,

Line 99 -“CapLight = encapsulation in 99 the presence of light; ExtLight = extract in the presence of light” - what temperature of storage?

Line 114- “(4 °C x 25 °C)” - maybe “4 °C, 25 °C” or “4 °C and 25 °C” will be more appropriate

Figure 1 - in my opinion Figure 1 is redundant , because  all this data  are in Table 1

Materials and Methods

Models of apparatus should be given for every analyses

Line 220 -it's not clear what "at a frequency of 40%" means, - there should be an explanation

Line 234 - How the amount of phenolic compounds inside and on the surface of the capsules was measured for EE? At work Selamat et al. hexan was used for extraction of tocopherols, but I don't think it is a good method for phenolics. It must be described more precisely.

Lines 251-252 - “Cap4 = 251 encapsulation at 4 °C; Ext4 = extract at 4 °C;” -with or without presence of light?

Lines 202 and 210 - numbers of references 29 and 30 should be interchanged because authors cited  Moser [30] first and Obon [29] second 

References

Line 335 - should be year 2013 instead 2012

Author Response

  1. The title promises more than the manuscript actually contains. There is one extraction method and one microencapsulation method. The authors focused more on the stability of the extract during storage under different conditions. I suggest changing the title to a more appropriate one.

The change was performed in the manuscript.

  1. Microencapsulation consists of closing the biologically active compound in the biopolymer, so that it is protected from light, oxygen, water or other environmental factors. The advantage of microencapsulation lies in that the core material is completely coated and isolated from the external environment. Did the authors have the opportunity to verify that the coating material actually coated the particles of the extract? What is the opinion of the authors on the distribution of phenolic compounds in the microcapsule and how they determined the content of phenolic compounds inside and outside the capsule to determine EE?

Our research group has worked with this technique and in all the studies the coating material actually coated the extract particles. To calculate the EE, we carefully followed a method reported in the literature (Castro et al. [33]), in which the procedure involves the disintegration and release of the compound inside the capsule, so we believe that this analysis reports the real efficiency value.

  1. Line 26 – TA and TP – explanation (total anthocyanins and total phenolics) should be given in first use

The explanation was added in the manuscript

  1. Line 82 -GX -explanation (xanthan gum) should be given in first use

The explanation was added in the manuscript

  1. Line 98 -“d=days” should be removed because in table the word “days” is used,

The word (d=days) was removed in the manuscript.

  1. Line 99 -“CapLight = encapsulation in 99 the presence of light; ExtLight = extract in the presence of light” - what temperature of storage?

Was at 25 °C, the information was performed in the manuscript.

  1. Line 114- “(4 °C x 25 °C)” - maybe “4 °C, 25 °C” or “4 °C and 25 °C” will be more appropriate

The change was performed in the manuscript.

  1. Figure 1 - in my opinion Figure 1 is redundant, because all this data are in Table 1

Thank you for your suggestion, but we would like to reiterate that figure 1 differs from table 1, as it aims to evaluate the microencapsulation process, so only the results of the capsules are shown in order to compare.

  1. Materials and Methods

Models of apparatus should be given for every analyses Line 220 -it's not clear what "at a frequency of 40%" means, - there should be an explanation

The information has been corrected; the correct figure is amplitude.

  1. Line 234 - How the amount of phenolic compounds inside and on the surface of the capsules was measured for EE? At work Selamat et al. hexan was used for extraction of tocopherols, but I don't think it is a good method for phenolics. It must be described more precisely.

Thank you for your comment, the reference was incorrect, we apologize, the information has been corrected in the text.

  1. Lines 251-252 - “Cap4 = 251 encapsulation at 4 °C; Ext4 = extract at 4 °C;” -with or without presence of light?

Without light presence, the information was corrected.

  1. Lines 202 and 210 - numbers of references 29 and 30 should be interchanged because authors cited Moser [30] first and Obon [29] second

Reference numbers 29 and 30 were checked.

  1. References

Line 335 - should be year 2013 instead 2012
The year was changed to 2013.